

# Electrical detection of the Majorana fusion rule for chiral edge vortices in a topological superconductor

C. W. J. Beenakker, A. Grabsch and Y. Herasymenko

Instituut-Lorentz, Universiteit Leiden, P.O. Box 9506, 2300 RA Leiden, The Netherlands

## Abstract

Majorana zero-modes bound to vortices in a topological superconductor have a non-Abelian exchange statistics expressed by a non-deterministic fusion rule: When two vortices merge they *may* or they *may not* produce an unpaired fermion with equal probability. Building on a recent proposal to inject edge vortices in a chiral mode by means of a Josephson junction, we show how the fusion rule manifests itself in an electrical measurement. A $2\pi$ phase shift at a pair of Josephson junctions creates a topological qubit in a state of even-even fermion parity, which is transformed by the chiral motion of the edge vortices into an equal-weight superposition of even-even and odd-odd fermion parity. Fusion of the edge vortices at a second pair of Josephson junctions results in a correlated charge transfer of zero or one electron per cycle, such that the current at each junction exhibits shot noise, but the difference of the currents is nearly noiseless.

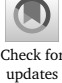
doi:10.21468/SciPostPhys.6.2.022

## 1  Introduction

Vortices in a two-dimensional topological superconductor contain a midgap state, or *zero-mode*, that can be used to store quantum mechanical information in a nonlocal way, protected from local sources of decoherence [1–5]. The qubit degree of freedom is the fermion parity of any two widely separated vortices, which may or may not share an unpaired electron or hole (a fermionic quasiparticle) in the condensate of Cooper pairs. The pairwise exchange, or *braiding*, of vortices is a unitary transformation which can serve as a building block for a quantum computation [6,7]. The merging, or *fusion*, of two vortices is the read-out operation [8]: The qubit is in the state $|1\rangle$ or $|0\rangle$ depending on whether or not the vortices leave behind a unpaired fermion. The fact that braiding operations do not commute, referred to as *non-Abelian statistics*, goes hand-in-hand with the fact that the fusion outcome is non-deterministic. As illustrated in Fig. 1, the fusion of two vortices $\sigma$ produces a quantum superposition of states $\psi$ and $\mathbb{I}$ with and without a quasiparticle excitation. This is the Majorana fusion rule[1] of non-Abelian anyons, symbolically written as $\sigma \otimes \sigma = \psi \oplus \mathbb{I}$.

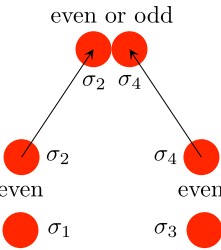

Figure 1: Schematic illustration of the fusion rule $\sigma_2 \otimes \sigma_4 = \psi \oplus \mathbb{I}$ of Majorana zero-modes (red dots, labeled $\sigma_n$). Pairs of zero-modes may or may not share a quasiparticle. In the former case the fermion parity is "odd" (indicated by $\psi$), in the latter case it is "even" (indicated by $\mathbb{I}$). The overall fermion parity is conserved, so if the fusion of $\sigma_2$ and $\sigma_4$ leaves behind a quasiparticle, then the fusion of $\sigma_1$ and $\sigma_3$ must also produce a quasiparticle.

Neither the braiding nor the fusion of vortices has been realized in the laboratory. This has motivated a variety of theoretical proposals for methods to demonstrate the appearance of non-Abelian anyons in a topological superconductor [10–14]. The obstacle that these proposals seek to remove, is the need to physically move the zero-modes around. Ref. 15 proposes an alternative approach: Substitute immobile bulk vortices for mobile edge vortices. In that paper

---

[1]Because of a mapping onto the Ising model, the term "Ising fusion rule" is also used.

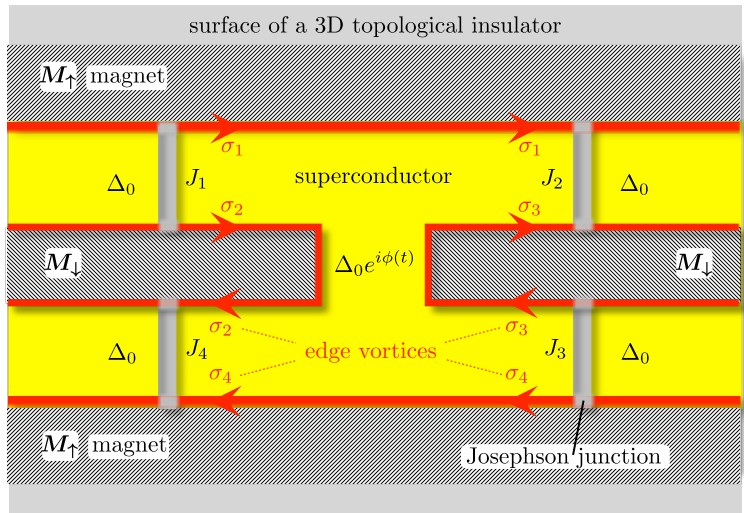

Figure 2: Geometry to create and fuse two pairs of edge vortices in a topological insulator/magnetic insulator/superconductor heterostructure. The edge vortices are created at Josephson junctions $J_1$ and $J_3$, by a $2\pi$ increment of the superconducting phase $\phi(t)$ on the central superconducting island. Each edge vortex contains a Majorana zero-mode and two zero-modes define a fermion parity qubit. The initial state $|J_1 J_3\rangle = |00\rangle$ has even-even fermion parity. When the edge vortices fuse at Josephson junctions $J_2$ and $J_4$ the final state $|J_2 J_4\rangle = (|00\rangle + i|11\rangle)/\sqrt{2}$ is in an equal-weight superposition of even-even and odd-odd parity states.

the braiding of vortices was considered. Here we turn to the fusion of edge vortices, in order to demonstrate the Majorana fusion rule.

Edge vortices are $\pi$-phase domain walls for Majorana fermions propagating along the edge of a topological superconductor [16]. Edge vortices may appear stochastically from quantum phase slips at a Josephson junction [17–19], but for our purpose we use the *deterministic* injector of Ref. 15: A voltage pulse $V(t)$ of integrated magnitude $\int V(t)dt = h/2e$ applied over a Josephson junction injects an edge vortex at each end of the junction. The injection happens when the phase difference $\phi$ of the superconducting pair potential crosses $\pi$. At $\phi = \pi$ the effective gap $\Delta_0 \cos(\phi/2)$ in the junction changes sign [20]. By the same mechanism that is operative in the Kitaev chain [21], the gap inversion creates a zero-mode at each end of the junction, which then propagates away from the junction along the edge mode. The edge modes are chiral, meaning that the motion is in a single direction only. For our purpose we need that the propagation is in the same direction along both edges connected by a Josephson junction. The geometry of Fig. 2 shows one way to achieve this using a topological insulator/magnetic insulator/superconductor heterostructure [22,23]. (In Fig. 3 we show an alternative realization using a Chern insulator/superconductor heterostructure [24,25].)

In the next section 2 we describe the way in which the fusion process shown schematically in Fig. 1 can be implemented in the structure of Figs. 2 and 3. In the subsequent sections 3 and 4 we present an explicit calculation of the fermion parity of the final state, to demonstrate the equal-weight superposition of even and odd fermion parity implied by the Majorana fusion rule. Sec. 5 addresses an electrical signature of the fusion process: The sum $I_L + I_R$ of the currents at the two ends of the structure shows shot noise, because of the nondeterministic nature of the fusion process, but the difference $I_L - I_R$ is nearly noiseless, because of the correlated fermion parity. We conclude in Sec. 6.

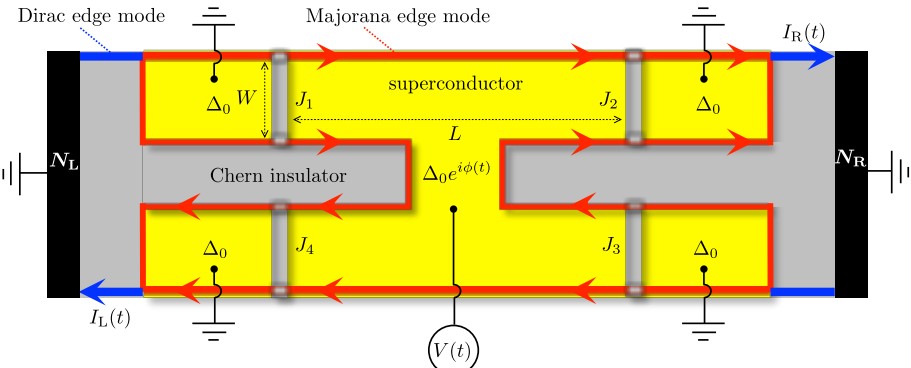

Figure 3: Same as Fig. 2, but now in a Chern insulator/superconductor heterostructure with normal metal contacts ($N_L$, $N_R$) to detect the charge produced upon fusion of the edge vortices. An integrated voltage pulse $\int V(t)dt = h/2e$ induces a $2\pi$ phase shift over the four Josephson junctions $J_1, J_2, J_3, J_4$, which results in a current pulse $I_L(t)$, $I_R(t)$ into the left and right contact. While $I_L$ and $I_R$ separately, as well as the sum $I_L + I_R$, exhibit shot noise, the difference $I_L - I_R$ becomes exactly noiseless for identical junctions $J_1$ and $J_3$.

## 2 Edge vortex injection and fusion in a four-terminal Josephson junction

The geometry of Fig. 2, with four incoming and four outgoing Majorana edge modes was introduced in Ref. 26 and studied recently in Refs. 27–29. Those earlier works considered the injection of *fermions*: electrons and holes injected into the Majorana edge modes from a normal metal contact. Here instead we consider the injection of *vortices*: $\pi$-phase domain walls injected into the edge modes by a Josephson junction. The injection happens in response to a voltage pulse $\int V(t)dt = h/2e$, which advances by $2\pi$ the phase $\phi(t)$ of the pair potential $\Delta_0 e^{i\phi}$. (Alternatively, an $h/2e$ flux bias achieves the same.) If the width $W$ of the Josephson junction is large compared to the superconducting coherence length $\xi_0 = \hbar v_F/\Delta_0$, the injection happens in a short time interval $t_\phi = (\xi_0/W)(\Delta t/2\pi)$ around $\phi(t) = \pi$, short compared the duration $\Delta t$ of the voltage pulse [15].[2]

The edge vortices $\sigma_n$ are anyons with a non-Abelian exchange statistics encoded in the Clifford algebra of Majorana operators $\gamma_n$,

$$\gamma_n\gamma_m + \gamma_m\gamma_n = \delta_{nm}. \tag{2.1}$$

Each edge vortex has a zero-mode and two zero-modes $n, m$ encode a qubit degree of freedom in the fermion parity $P_{nm} = 2i\gamma_n\gamma_m$ with eigenvalues $\pm 1$. Provided the vortices are non-overlapping, the qubit is protected from local sources of decoherence.

In the four-terminal Josephson junction of Fig. 2, one pair of edge vortices $\sigma_1, \sigma_2$ is injected at Josephson junction $J_1$ and a second pair $\sigma_3, \sigma_4$ is injected at Josephson junction $J_3$. Because the voltage pulse cannot create an unpaired fermion, the edge vortices are injected in a state $|\Psi\rangle$ of even fermion parity, $P_{12}|\Psi\rangle = |\Psi\rangle = P_{34}|\Psi\rangle$. Edge vortices $\sigma_1$ and $\sigma_3$ are fused at Josephson junction $J_2$ and vortices $\sigma_2$ and $\sigma_4$ are fused at junction $J_4$. The expectation value

---

[2]This separation of time scales $t_\phi/\Delta t \simeq \xi_0/W \ll 1$ is why it is meaningful to distinguish the injection of vortices from the injection of fermions, since a Majorana fermion in an edge mode is equivalent to a pair of overlapping edge vortices.

of the fermion parity upon fusion vanishes,

$$\langle\Psi|P_{13}|\Psi\rangle = \langle\Psi|P_{12}P_{13}P_{12}|\Psi\rangle = -\langle\Psi|P_{13}P_{12}^2|\Psi\rangle = -\langle\Psi|P_{13}|\Psi\rangle$$
$$\Rightarrow \langle\Psi|P_{13}|\Psi\rangle = 0, \tag{2.2}$$

and similarly $\langle\Psi|P_{24}|\Psi\rangle = 0$. So the fusion of edge vortices at $J_2$ and $J_3$ leaves the edge modes in an equal weight superposition of odd and even fermion parity. This presence of multiple fusion channels is a defining property of non-Abelian anyons [3–5].

Because the overall fermion parity is conserved, the fusion outcomes at $J_2$ and $J_3$ must have the same fermion parity — either even-even or odd-odd. In the next two sections we present an explicit calculation of the fermion parity, to demonstrate that an $h/2e$ voltage pulse produces a superposition of even-even and odd-odd fermion parity states with identical probabilities $P_{00}$ and $P_{11} = 1 - P_{00}$.

## 3 Scattering formula for the fermion parity

### 3.1 Construction of the fermion parity operator

We focus on the geometry of Fig. 3, with incoming and outgoing modes in the left lead (labeled L) and in the right lead (R). We seek the expectation value

$$\rho_\pi \equiv \left\langle e^{i\pi\mathcal{N}} \right\rangle = P_{00} - P_{11}, \tag{3.1}$$

of the fermion parity operator $e^{i\pi\mathcal{N}}$, with $\mathcal{N}$ the particle number operator of outgoing modes in one of the two leads. We will take the left lead for definiteness. In terms of the annilation operators $b_n(E)$ of outgoing modes $n$ at excitation energy $E > 0$ this operator takes the form

$$\mathcal{N} = \sum_{n\in\text{L}}\sum_{E>0} b_n^\dagger(E)b_n(E), \tag{3.2}$$

where we have discretized the energy. In the continuum limit $\sum_E \mapsto \int dE/2\pi$ and the Kronecker delta becomes a Dirac delta function, $\delta_{EE'} \mapsto 2\pi\delta(E - E')$.

Incoming and outgoing modes are related by a unitary scattering matrix,

$$b_n(E) = \sum_{m,E'} S_{nm}(E,E')a_m(E'), \tag{3.3}$$

$$\sum_{n'',E''} S_{n''n}^*(E'',E)S_{n''m}(E'',E') = \delta_{nm}\delta_{EE'}. \tag{3.4}$$

Note that the sums in these two equations run over positive and negative energies. Particle-hole symmetry relates

$$S_{nm}(-E,-E') = S_{nm}^*(E,E'). \tag{3.5}$$

We write Eq. (3.3) more compactly as $\boldsymbol{b} = \boldsymbol{S}\cdot\boldsymbol{a}$, collecting the mode and energy variables in vectors $\boldsymbol{a}$ and $\boldsymbol{b}$. The unitarity relation (3.4) is then written as $\boldsymbol{S}^\dagger\boldsymbol{S} = 1$. In terms of a projection operator $\mathcal{P}_\text{L}$ onto modes in lead L, and a projection operator $\mathcal{P}_+$ onto positive energies, the combination of Eqs. (3.2) and (3.3) reads

$$\mathcal{N} = \boldsymbol{a}^\dagger\cdot\boldsymbol{M}\cdot\boldsymbol{a}, \ \ \boldsymbol{M} = \boldsymbol{S}^\dagger\mathcal{P}_\text{L}\mathcal{P}_+\boldsymbol{S}. \tag{3.6}$$

The expectation value $\langle\cdots\rangle = \text{Tr}(\rho_\text{eq}\cdots)$ is with respect to an equilibrium distribution of the incoming modes,

$$\rho_\text{eq} \propto \exp\left(-\beta\sum_n\sum_{E>0} Ea_n^\dagger(E)a_n(E)\right). \tag{3.7}$$

We denote $\beta = 1/k_{\mathrm{B}}T$ and have omitted the normalization constant (fixed by $\mathrm{Tr}\,\rho_{\mathrm{eq}} = 1$).

The combination of particle-hole symmetry,

$$a_n^\dagger(E) = a_n(-E), \tag{3.8}$$

with anticommutation,

$$\{a_n^\dagger(E), a_m(E')\} = \delta_{nm}\delta_{EE'}, \tag{3.9}$$

allows us to extend the sum $\sum_{E>0}$ in Eq. (3.7) to a sum over positive and negative energies,

$$\rho_{\mathrm{eq}} \propto \exp\!\left(-\tfrac{1}{2}\beta \sum_{n,E} E a_n^\dagger(E) a_n(E)\right) \equiv e^{-\frac{1}{2}\beta \boldsymbol{a}^\dagger \cdot \boldsymbol{E} \cdot \boldsymbol{a}}. \tag{3.10}$$

In the second equation we introduced the diagonal operator $\boldsymbol{E}_{nm}(E, E') = E\delta_{nm}\delta_{EE'}$.

With this notation the average fermion parity is given by the ratio of two operator traces,

$$\rho_\pi = \frac{\mathrm{Tr}\left(e^{-\frac{1}{2}\beta \boldsymbol{a}^\dagger \cdot \boldsymbol{E} \cdot \boldsymbol{a}} e^{i\pi \boldsymbol{a}^\dagger \cdot \boldsymbol{M} \cdot \boldsymbol{a}}\right)}{\mathrm{Tr}\, e^{-\frac{1}{2}\beta \boldsymbol{a}^\dagger \cdot \boldsymbol{E} \cdot \boldsymbol{a}}}. \tag{3.11}$$

## 3.2 Klich formula for particle-hole conjugate Majorana operators

Fermionic operator traces of the form (3.11) have been studied by Klich and collaborators [30–32]. For Dirac fermion creation and annihilation operators $\boldsymbol{d}^\dagger, \boldsymbol{d}$ one has the simple expression [30]

$$\mathrm{Tr} \prod_k e^{\boldsymbol{d}^\dagger \cdot \boldsymbol{O}_k \cdot \boldsymbol{d}} = \mathrm{Det}\left(1 + \prod_k e^{\boldsymbol{O}_k}\right). \tag{3.12}$$

The answer is different for self-conjugate Majorana operators $\boldsymbol{\gamma} = \boldsymbol{\gamma}^\dagger$, with anticommutator $\{\gamma_n, \gamma_m\} = \delta_{nm}$, when one has instead [32]

$$\left[\mathrm{Tr} \prod_k e^{\boldsymbol{\gamma}^\dagger \cdot \boldsymbol{O}_k \cdot \boldsymbol{\gamma}}\right]^2 = e^{\sum_k \mathrm{Tr}\,\boldsymbol{O}_k}\, \mathrm{Det}\left(1 + \prod_k e^{\boldsymbol{O}_k - \boldsymbol{O}_k^{\mathrm{T}}}\right). \tag{3.13}$$

(The superscript T indicates the transpose of the matrix.)

The Majorana fermion modes in the topological superconductor are not self-conjugate, instead creation and annihilation operators $\boldsymbol{a}^\dagger, \boldsymbol{a}$ are related by the particle-hole symmetry relation (3.8). In view of Eq. (3.9) this implies that annihilation operators at energies $\pm E$ fail to anticommute:

$$\{a_n(E), a_m(-E')\} = \delta_{nm}\delta_{EE'}. \tag{3.14}$$

This unusual anticommutator expresses the Majorana nature of Bogoliubov quasiparticles [33].

To arrive at the analogue of Eq. (3.13) for particle-hole conjugate Majorana operators we rewrite the bilinear form $\boldsymbol{a}^\dagger \cdot \boldsymbol{O} \cdot \boldsymbol{a}$ such that the $\boldsymbol{a}, \boldsymbol{a}^\dagger$ operators appear only at positive energies:

$$\boldsymbol{a}^\dagger \cdot \boldsymbol{O} \cdot \boldsymbol{a} = \sum_{n,m}\sum_{E,E'} a_n^\dagger(E) O_{nm}(E,E') a_m(E') = \sum_{n,m}\sum_{E,E'>0} \begin{pmatrix} a_n^\dagger(E) \\ a_n(E) \end{pmatrix} \mathcal{O}_{nm}(E,E') \begin{pmatrix} a_m(E') \\ a_m^\dagger(E') \end{pmatrix}. \tag{3.15}$$

The matrix $\mathcal{O}$ imposes on $\boldsymbol{O}$ a $2 \times 2$ block structure,

$$\mathcal{O} = \begin{pmatrix} \boldsymbol{O}_{++} & \boldsymbol{O}_{+-} \\ \boldsymbol{O}_{-+} & \boldsymbol{O}_{--} \end{pmatrix}, \tag{3.16}$$

to encode the sign of the energy variables:

$$(\boldsymbol{O}_{ss'})_{nm}(E,'E') = O_{nm}(sE, s'E') \text{ for } s,s' \in \{+,-\} \text{ and } E,E' > 0. \tag{3.17}$$

We introduce the $2 \times 2$ Pauli matrix $\sigma_x$ that acts on the block structure of $\mathcal{O}$ and define the generalized antisymmetrization

$$\mathcal{O}^{\mathrm{A}} = \tfrac{1}{2}\mathcal{O} - \tfrac{1}{2}\sigma_x \mathcal{O}^{\mathrm{T}}\sigma_x = \frac{1}{2}\begin{pmatrix} \boldsymbol{O}_{++} - \boldsymbol{O}_{--}^{\mathrm{T}} & \boldsymbol{O}_{+-} - \boldsymbol{O}_{+-}^{\mathrm{T}} \\ \boldsymbol{O}_{-+} - \boldsymbol{O}_{-+}^{\mathrm{T}} & \boldsymbol{O}_{--} - \boldsymbol{O}_{++}^{\mathrm{T}} \end{pmatrix}. \tag{3.18}$$

Only $\mathcal{O}^{\mathrm{A}}$ and $\mathrm{Tr}\,\mathcal{O} = \mathrm{Tr}\,\boldsymbol{O}$ contribute to the Majorana fermion operator trace,

$$\left[ \mathrm{Tr}\prod_k e^{\boldsymbol{a}^\dagger \cdot \boldsymbol{O}_k \cdot \boldsymbol{a}} \right]^2 = e^{\sum_k \mathrm{Tr}\,\boldsymbol{O}_k}\,\mathrm{Det}\left( 1 + \prod_k e^{2\mathcal{O}_k^{\mathrm{A}}} \right), \tag{3.19}$$

see App. B. Eq. (3.19) is the desired analogue of Eq. (3.13) for particle-hole conjugate Majorana operators.

## 3.3 Fermion parity as the determinant of a scattering matrix product

For the average fermion parity $\rho_\pi$ we apply Eq. (3.19) to the ratio of operator traces (3.11). We start from the block decomposition of $\boldsymbol{E}, \boldsymbol{S}$, and $\boldsymbol{M} = \boldsymbol{S}^\dagger \mathcal{P}_{\mathrm{L}}\mathcal{P}_+ \boldsymbol{S}$,

$$\mathcal{E} = \begin{pmatrix} \boldsymbol{E} & 0 \\ 0 & -\boldsymbol{E} \end{pmatrix} = \boldsymbol{E}\sigma_z, \quad \mathcal{S} = \begin{pmatrix} \boldsymbol{S}_{++} & \boldsymbol{S}_{+-} \\ \boldsymbol{S}_{-+} & \boldsymbol{S}_{--} \end{pmatrix},$$
$$\mathcal{M} = \tfrac{1}{2}\mathcal{S}^\dagger \mathcal{P}_{\mathrm{L}}(\sigma_0 + \sigma_z)\mathcal{S}. \tag{3.20}$$

In the equation for $\mathcal{M}$ we substituted $\mathcal{P}_+ = \tfrac{1}{2}(\sigma_0 + \sigma_z)$, with $\sigma_0$ the $2 \times 2$ unit matrix.

The antisymmetrization of $\mathcal{E}$ is simple,

$$\mathcal{E}^{\mathrm{A}} \equiv \tfrac{1}{2}\mathcal{E} - \tfrac{1}{2}\sigma_x \mathcal{E}^{\mathrm{T}}\sigma_x = \boldsymbol{E}\sigma_z. \tag{3.21}$$

For the antisymmetrization of $\mathcal{M}$ we note that Eq. (3.5) implies $\sigma_x \mathcal{S}\sigma_x = \mathcal{S}^*$, hence

$$\sigma_x \mathcal{S}^{\mathrm{T}}\sigma_x = \mathcal{S}^\dagger \Rightarrow \mathcal{M}^{\mathrm{A}} = \tfrac{1}{2}\mathcal{S}^\dagger \mathcal{P}_{\mathrm{L}}\sigma_z \mathcal{S}. \tag{3.22}$$

We thus arrive at

$$\rho_\pi^2 = e^{i\pi\,\mathrm{Tr}\,\boldsymbol{M}} \frac{\mathrm{Det}(1 + e^{-\beta E\sigma_z}e^{i\pi\mathcal{S}^\dagger \mathcal{P}_{\mathrm{L}}\sigma_z \mathcal{S}})}{\mathrm{Det}(1 + e^{-\beta E\sigma_z})}. \tag{3.23}$$

The ratio of determinants is equivalent to a single determinant,

$$\rho_\pi^2 = e^{i\pi\,\mathrm{Tr}\,\boldsymbol{M}}\,\mathrm{Det}\left( 1 - \mathcal{F} + \mathcal{F}e^{i\pi\mathcal{S}^\dagger \mathcal{P}_{\mathrm{L}}\sigma_z \mathcal{S}} \right),$$
$$\mathcal{F} = (1 + e^{\beta E\sigma_z})^{-1}, \quad 1 - \mathcal{F} = (1 + e^{-\beta E\sigma_z})^{-1}. \tag{3.24}$$

To proceed we first rewrite the exponent of the trace of $\boldsymbol{M}$ as a determinant,

$$e^{i\pi\,\mathrm{Tr}\,\boldsymbol{M}} = e^{i\pi\,\mathrm{Tr}\,\mathcal{P}_{\mathrm{L}}\mathcal{P}_+} \tag{3.25a}$$

$$= \mathrm{Det}\,[-\sigma_z]_{\mathrm{LL}} = \mathrm{Det}\,[\sigma_z]_{\mathrm{LL}} \text{ with } \sigma_z \equiv 2\mathcal{P}_+ - 1, \tag{3.25b}$$

$$= \mathrm{Det}\,[-\tau_z]_{++} = \mathrm{Det}\,[\tau_z]_{++} \text{ with } \tau_z \equiv 2\mathcal{P}_{\mathrm{L}} - 1. \tag{3.25c}$$

The notation $[\cdots]_{\mathrm{LL}}$ indicates a projection onto mode indices in the left lead, and $[\cdots]_{++}$ indicates a projection onto positive energies.

We then evaluate the exponent of the scattering matrix product,

$$e^{i\xi \mathcal{S}^\dagger \mathcal{P}_L \sigma_z \mathcal{S}} = \sigma_0 + i(\sin \xi)\mathcal{S}^\dagger \mathcal{P}_L \sigma_z \mathcal{S} + (\cos \xi - 1)\mathcal{S}^\dagger \mathcal{P}_L \mathcal{S},$$

$$\Rightarrow e^{i\pi \mathcal{S}^\dagger \mathcal{P}_L \sigma_z \mathcal{S}} = \sigma_0 - 2\mathcal{S}^\dagger \mathcal{P}_L \mathcal{S}, \tag{3.26}$$

since $(\mathcal{S}^\dagger \mathcal{P}_L \sigma_z \mathcal{S})^{2n} = \mathcal{S}^\dagger \mathcal{P}_L \mathcal{S}$ and $(\mathcal{S}^\dagger \mathcal{P}_L \sigma_z \mathcal{S})^{2n-1} = \mathcal{S}^\dagger \mathcal{P}_L \sigma_z \mathcal{S}$, for $n = 1, 2, 3, \ldots$. It follows that

$$\rho_\pi^2 = e^{i\pi \operatorname{Tr} M} \operatorname{Det}\left(1 - 2\mathcal{F}\mathcal{S}^\dagger \mathcal{P}_L \mathcal{S}\right) \tag{3.27a}$$

$$= e^{i\pi \operatorname{Tr} M} \operatorname{Det}\left(1 - 2\mathcal{P}_L \mathcal{S}\mathcal{F}\mathcal{S}^\dagger\right) \tag{3.27b}$$

$$= e^{i\pi \operatorname{Tr} M} \operatorname{Det}\left[1 - 2\mathcal{S}\mathcal{F}\mathcal{S}^\dagger\right]_{LL} \tag{3.27c}$$

$$= \operatorname{Det}\left[\sigma_z\right]_{LL} \operatorname{Det}\left[\mathcal{S}(1 - 2\mathcal{F})\mathcal{S}^\dagger\right]_{LL} \tag{3.27d}$$

$$= \operatorname{Det}\left[\sigma_z \mathcal{S} \tanh(\tfrac{1}{2}\beta \mathcal{E})\mathcal{S}^\dagger\right]_{LL}. \tag{3.27e}$$

In Eq. (3.27b) we used the Sylvester identity $\operatorname{Det}(1 - AB) = \operatorname{Det}(1 - BA)$, in Eq. (3.27c) we used $\operatorname{Det}(1 - \mathcal{P}_L A) = \operatorname{Det}[1 - A]_{LL}$, in Eq. (3.27d) we used $\mathcal{S}\mathcal{S}^\dagger = 1$, and in (3.27e) we used that $\operatorname{Det}[A]_{LL}\operatorname{Det}[B]_{LL} = \operatorname{Det}[AB]_{LL}$ if $A$ or $B$ commutes with $\mathcal{P}_L$.

In what follows we restrict ourselves to zero temperature, when $\mathcal{F} \mapsto \mathcal{P}_-$ projects onto negative energies and $\tanh(\tfrac{1}{2}\beta \mathcal{E}) \mapsto \sigma_z$. Eq. (3.27e) then reduces to

$$\rho_\pi^2 = \operatorname{Det}\left[\sigma_z \mathcal{S} \sigma_z \mathcal{S}^\dagger\right]_{LL}, \tag{3.28}$$

the determinant of a scattering matrix product projected onto mode indices in the left lead. An alternative projection onto positive energies is possible:

$$\rho_\pi^2 = e^{i\pi \operatorname{Tr} M} \operatorname{Det}\left(1 - 2\mathcal{P}_- \mathcal{S}^\dagger \mathcal{P}_L \mathcal{S}\right) \tag{3.29a}$$

$$= e^{i\pi \operatorname{Tr} M} \operatorname{Det}\left(1 - 2\mathcal{P}_+ \mathcal{S}^\dagger \mathcal{P}_L \mathcal{S}\right) \tag{3.29b}$$

$$= \operatorname{Det}\left[-\tau_z\right]_{++} \operatorname{Det}\left[\mathcal{S}^\dagger(1 - 2\mathcal{P}_L)\mathcal{S}\right]_{++}, \tag{3.29c}$$

(In Eq. (3.29b) we used particle-hole symmetry, $\mathcal{S} = \sigma_x \mathcal{S}^* \sigma_x$, and $\sigma_x \mathcal{P}_- \sigma_x = \mathcal{P}_+$.) Because $\tau_z$ commutes with $\mathcal{P}_+$, Eq. (3.29c) may be combined into a a single determinant,

$$\rho_\pi^2 = \operatorname{Det}\left[\tau_z \mathcal{S}^\dagger \tau_z \mathcal{S}\right]_{++}. \tag{3.30}$$

Equations (3.28) and (3.30) express the average fermion parity of a scattering state as the determinant of a product of scattering matrices projected onto a submatrix in mode space, Eq. (3.28), or in energy space, Eq. (3.30).[3] Both equations give the square $\rho_\pi^2$ rather than $\rho_\pi$ itself. Since we wish to show that $\rho_\pi = 0$, that is not a limitation for the present study.

## 3.4 Simplification in the adiabatic regime

The energy dependence of the scattering matrix is characterized by the inverse of two time scales of the Josephson junction: the dwell time $\tau_{\text{dwell}} \simeq L/v_F$ in the superconducting island and the characteristic time scale

$$t_\phi = (\xi_0/W)(d\phi/dt)^{-1} \tag{3.31}$$

for the variation of the superconducting phase shift. (The time $t_\phi$ is the "vortex injection time" $t_{\text{inj}}$ of Ref. 15.) While $S(E, E')$ depends on the average energy $\bar{E} = (E + E')/2$ on the scale $1/\tau_{\text{dwell}}$, it depends on the energy difference $\delta E = E - E'$ on the scale $1/\tau_\phi$.

---

[3]To avoid a possible confusion we note that, because of the projection, the product rule $\operatorname{Det}(AB) = (\operatorname{Det} A)(\operatorname{Det} B)$ cannot be applied to $\operatorname{Det}[AB]_{++}$ or $\operatorname{Det}[AB]_{LL}$, unless $A$ or $B$ commutes with the projector.

In the adiabatic regime $\tau_{\text{dwell}} \ll \tau_\phi$ the scattering matrix $S(E, E')$ for $\bar{E} \lesssim 1/\tau_\phi \ll 1/\tau_{\text{dwell}}$ is only a function of $\delta E$,

$$S(E, E') = \int_{-\infty}^{\infty} dt\, e^{i(E-E')t} S_{\text{F}}(t) + \mathcal{O}(\tau_{\text{dwell}}/\tau_\phi). \tag{3.32}$$

The unitary matrix $S_{\text{F}}(t)$ is the "frozen" scattering matrix at the Fermi level, calculated for a fixed value $\phi \equiv \phi(t)$ of the superconducting phase.

The fermion parity determinant can be simplified in the adiabatic regime, because only energies within $1/\tau_\phi$ from the Fermi level contribute. This is most easily seen from Eq. (3.28), which is the determinant of the scattering matrix product $\Omega = \sigma_z \mathcal{S} \sigma_z \mathcal{S}^\dagger$, projected onto the left lead. A matrix element of $\Omega$,

$$\Omega_{nm}(E, E') = (\text{sign}\, E) \sum_{n', E''} (\text{sign}\, E'') S_{nn'}(E, E'') S_{mn'}^*(E', E'') \tag{3.33}$$

is only nonzero for $|E - E'| \lesssim 1/\tau_\phi$. Moreover, $\Omega_{nm}(E, E') \approx \delta_{nm}\delta_{EE'}$ for $|E| \gtrsim 1/\tau_\phi$. Hence the determinant of $\Omega$ is fully determined by energies in the range $-1/\tau_\phi \lesssim E, E' \lesssim \tau_\phi$, where $S(E, E')$ may be approximated by the frozen scattering matrix (3.32).

For computational purposes it is more convenient to rewrite the determinant (3.28) in the form (3.30), because the scattering matrix product $\tau_z \mathcal{S} \tau_z \mathcal{S}^\dagger$ is a convolution in energy space when $S(E, E')$ is a function of $E - E'$. The convolution is readily evaluated in the time domain, resulting in an expression for the fermion parity

$$\rho_\pi^2 = \text{Det}\,[Q]_{++}, \tag{3.34}$$

in terms of the determinant of the projection onto $E, E' > 0$ of the matrix

$$Q(E, E') = \int_{-\infty}^{\infty} dt\, e^{i(E-E')t} Q(t), \quad Q(t) = \tau_z S_{\text{F}}^\dagger(t) \tau_z S_{\text{F}}(t). \tag{3.35}$$

In the next section we shall show how to evaluate this determinant.

## 4 Vanishing of the average fermion parity

We apply the formalism that we developed in Sec. 3 to the four-terminal Josephson junction of Sec. 2, in order to demonstrate that the $2\pi$ phase shift produces a state with an equal weight $P_{00} = P_{11}$ of even-even and odd-odd fermion parity in the left and right leads. We work in the adiabatic regime, when $\rho_\pi = P_{00} - P_{11}$ is given by Eqs. 3.34 and (3.35) in terms of the "frozen" scattering matrix $S_{\text{F}}(t)$, for a fixed phase $\phi(t)$.

### 4.1 Frozen scattering matrix of the Josephson junction

The frozen scattering matrix $S_{\text{F}} \in \text{SO}(4)$ is calculated in App. A, resulting in

$$S_{\text{F}} = \begin{pmatrix} e^{-i\alpha_4 \nu_y} & 0 \\ 0 & e^{-i\alpha_2 \nu_y} \end{pmatrix} \cdot \Pi \cdot \begin{pmatrix} e^{i\alpha_1 \nu_y} & 0 \\ 0 & e^{i\alpha_3 \nu_y} \end{pmatrix}, \quad \Pi = \begin{pmatrix} 0 & 0 & 1 & 0 \\ 0 & -1 & 0 & 0 \\ 1 & 0 & 0 & 0 \\ 0 & 0 & 0 & 1 \end{pmatrix}. \tag{4.1}$$

The Pauli matrix $\nu_y$ acts on the two Majorana modes in each lead. The scattering phase $\alpha_n$ depends on the superconducting phase difference $\phi$ through the relation [15]

$$\alpha_n = \arccos\left(\frac{\cos(\phi/2) + \tanh\beta_n}{1 + \cos(\phi/2)\tanh\beta_n}\right) \times \text{sign}(\phi), \quad \beta_n = \frac{W_n}{\xi_0}\cos(\phi/2). \tag{4.2}$$

A $2\pi$ increment of $\phi$ corresponds to a $\pi$ increment of $\alpha_n$, irrespective of the width $W_n$ of the Josephson junction or the superconducting coherence length $\xi_0 = \hbar v_F / \Delta_0$.

We need to evaluate the matrix product $\tau_z S_F^\dagger \tau_z S_F$, where the Pauli matrix

$$\tau_z = \begin{pmatrix} 1 & 0 & 0 & 0 \\ 0 & 1 & 0 & 0 \\ 0 & 0 & -1 & 0 \\ 0 & 0 & 0 & -1 \end{pmatrix} \tag{4.3}$$

is defined with respect to the block structure of modes in the left and right lead. Because of the identity

$$\Pi \tau_z \Pi = \begin{pmatrix} \nu_z & 0 \\ 0 & \nu_z \end{pmatrix}, \tag{4.4}$$

this matrix product is block-diagonal,

$$Q(t) = \tau_z S_F^\dagger(t) \tau_z S_F(t) = -\begin{pmatrix} \nu_z e^{2i\nu_y \alpha_1(t)} & 0 \\ 0 & \nu_z e^{2i\nu_y \alpha_3(t)} \end{pmatrix}, \tag{4.5}$$

independent of $\alpha_2$ and $\alpha_4$.

## 4.2 Reduction of the fermion parity to a Toeplitz determinant

Instead of taking a single $2\pi$ phase increment it is more convenient to assume a sequence of $2\pi$ phase shifts with period $\Delta t$. Then $\alpha_n(t)$ varies periodically in time with $\alpha_n(t+\Delta t) = \pi + \alpha_n(t)$. We Fourier transform to the energy domain,

$$
\begin{aligned}
\boldsymbol{T}_n(k,k') &= \frac{1}{\Delta t} \int_0^{\Delta t} dt\, e^{2\pi i (k-k')t/\Delta t} e^{2i\alpha_n(t)\nu_y}, \\
T_n(k,k') &= \frac{1}{\Delta t} \int_0^{\Delta t} dt\, e^{2\pi i (k-k')t/\Delta t} e^{2i\alpha_n(t)},
\end{aligned} \tag{4.6}
$$

and restrict $k, k' \in \{1, 2, 3, \ldots\}$ to positive integers. The infinite matrix $T_n(k,k')$ has constant diagonals, so it is a Toeplitz matrix. Eq. (3.30) becomes the product of Toeplitz determinants,

$$\rho_\pi^2 = (\text{Det } \boldsymbol{T}_1)(\text{Det } \boldsymbol{T}_3) = |\text{Det } T_1|^2 \, |\text{Det } T_3|^2. \tag{4.7}$$

The Toeplitz matrices $T_n$ are banded matrices which extend over a large number of order $W/\xi_0$ of diagonals around the main diagonal. This follows from the fact that the $\pi$ increment of $\alpha(t)$ happens in the time interval $t_\phi = (\xi_0/W)(\Delta t/2\pi)$ which is much shorter than $\Delta t$ for $\xi_0 \ll W$. The ratio $t_\phi/\Delta t$ governs the exponential decay of the Toeplitz matrix elements as one moves away from the main diagonal, according to

$$|T_n(k,k')| \simeq \exp(-c_{\text{decay}}|k - k'|), \quad c_{\text{decay}} = \frac{\pi^2 t_\phi}{\Delta t} = \frac{\pi \xi_0}{2W}. \tag{4.8}$$

## 4.3 Fisher-Hartwig asymptotics

In a general formulation, the function $b(\theta)$ defines the $K \times K$ Toeplitz matrix

$$B_K(k,k') = \int_0^{2\pi} e^{i(k-k')\theta} b(\theta) \frac{d\theta}{2\pi}, \quad k, k' \in \{1, 2, \ldots K\}. \tag{4.9}$$

If $b$ is smooth and nonvanishing on the unit circle $0 < \theta < 2\pi$, it has a well-defined winding number

$$\nu = \frac{1}{2\pi i} \int_0^{2\pi} \frac{b'(\theta)}{b(\theta)} \, d\theta. \tag{4.10}$$

The number $\nu$ may be non-integer, or even complex, if $b$ has a jump discontinuity at $\theta = 0$.

The Fisher-Hartwig asymptotics [34, 35] determines the large-$K$ limit of the determinant of $B_K$ from the decomposition $b(\theta) = b_0(\theta)e^{i\nu\theta}$, where $b_0$ has zero winding number. In the most general case the function $b_0$ may have (integrable) singularities, but if we assume it is smooth the asymptotics reads

$$\operatorname{Det} B_K \simeq \exp\left( \frac{K}{2\pi} \int_0^{2\pi} \ln b_0(\theta) \, d\theta \right) \times \begin{cases} K^{-\nu^2} & \text{for non-integer } \nu, \\ e^{-|\nu|c_{\text{decay}}K} & \text{for integer } \nu. \end{cases} \tag{4.11}$$

The coefficient $c_{\text{decay}}$ in the exponent is the decay rate $|B_K(k, k')| \simeq \exp(-c_{\text{decay}}|k - k'|)$ of the Toeplitz matrix elements as we move away from the diagonal.

Applied to $b(t) = e^{2i\alpha(t)}$, $\theta = 2\pi t/\Delta t$, we have $\nu = 1$, $b_0(t) = e^{2i\alpha(t)-2\pi it/\Delta t}$. The Toeplitz determinant

$$\operatorname{Det} B_K \simeq e^{-c_{\text{decay}}K} \exp\left( \frac{2iK}{\Delta t} \int_0^{\Delta t} \alpha(t) \, dt - i\pi K \right) \tag{4.12}$$

vanishes exponentially in the limit $K \to \infty$, with decay rate $c_{\text{decay}} = \pi\xi_0/W$ determined by the ratio of the superconducting coherence length $\xi_0$ and the width $W$ of the Josephson junction.

For the evaluation of the fermion parity, the band width $K/\Delta t$ is limited by the energy range $|\bar{E}| \lesssim 1/t_{\text{dwell}}$ where the dependence of the scattering matrix $S(E, E')$ on the average energy $\bar{E} = (E + E')/2$ may be neglected. We thus conclude that

$$|\rho_\pi| \simeq \exp(-2c_{\text{decay}}K) \simeq \exp\left( -\frac{2\pi\xi_0}{W} \frac{\Delta t}{t_{\text{dwell}}} \right) \simeq \exp\left( -\frac{4\pi^2 t_\phi}{t_{\text{dwell}}} \right), \tag{4.13}$$

which is exponentially small in the adiabatic regime $t_\phi \gg t_{\text{dwell}}$.

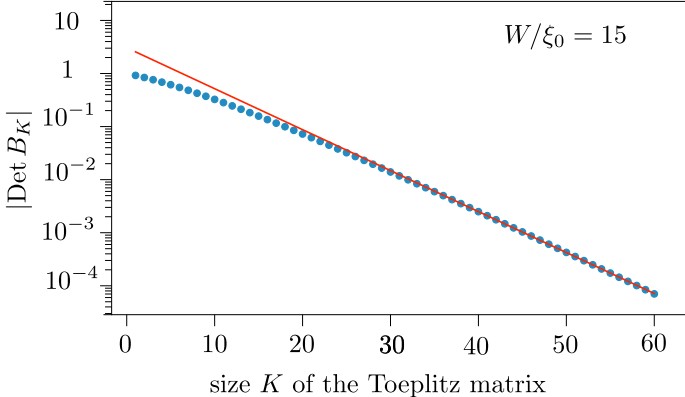

Figure 4: Decay of the Toeplitz determinant compared with the exponential decay expected from Eq. (4.12). The data points are calculated directly from Eq. (4.9) with $b(\theta) = \exp[2i\alpha(t)]$, $\theta \equiv \phi = 2\pi t/\Delta t$, given by Eq. (4.2) for $W/\xi_0 = 15$. The constant $c_{\text{decay}}$ was calculated separately from $|B_K(k, k')| \simeq \exp(-c_{\text{decay}}|k - k'|)$. The estimate $c_{\text{decay}} = \pi\xi_0/W$ is off by 15%.

# 5 Transferred charge

## 5.1 Average charge

The average charge $\langle Q_L \rangle$, $\langle Q_R \rangle$ transferred into the left or right lead during one $2\pi$ increment of $\phi$ is given, in the adiabatic regime, by the superconducting analogue of Brouwer's formula [36, 37]:

$$
\begin{aligned}
\langle Q_L \rangle &= \frac{ie}{4\pi} \int_{-\infty}^{\infty} dt \, \mathrm{Tr} \, S_F^\dagger(t) \begin{pmatrix} \nu_y & 0 \\ 0 & 0 \end{pmatrix} \frac{\partial}{\partial t} S_F(t), \\
\langle Q_R \rangle &= \frac{ie}{4\pi} \int_{-\infty}^{\infty} dt \, \mathrm{Tr} \, S_F^\dagger(t) \begin{pmatrix} 0 & 0 \\ 0 & \nu_y \end{pmatrix} \frac{\partial}{\partial t} S_F(t).
\end{aligned}
\tag{5.1}
$$

Substitution of Eq. (4.1) gives

$$
\begin{aligned}
\langle Q_L \rangle &= \frac{e}{2\pi} \int_{-\infty}^{\infty} dt \, \frac{d}{dt} \alpha_4(t), \\
\langle Q_R \rangle &= \frac{e}{2\pi} \int_{-\infty}^{\infty} dt \, \frac{d}{dt} \alpha_2(t).
\end{aligned}
\tag{5.2}
$$

Because both $\alpha_2$ and $\alpha_4$ increase by $\pi$ when $\phi$ is incremented by $2\pi$, see Eq. (4.2), we conclude that

$$
\langle Q_L \rangle = \langle Q_R \rangle = \frac{e}{2}.
\tag{5.3}
$$

While the average transferred charge per cycle is exactly $e/2$, the average particle number is close to but not exactly equal to $1/2$ — indicating that there is a small contribution from charge-neutral particle-hole pairs.[4]

## 5.2 Charge correlations

Fluctuations in the transferred charge are described by the second moments $\langle Q_L^2 \rangle$, $\langle Q_R^2 \rangle$, and $\langle Q_L Q_R \rangle$. Scattering matrix formulas for these correlators are derived in App. C. In the adiabatic regime one has

$$
\mathrm{var}(Q_L) \equiv \langle Q_L^2 \rangle - \langle Q_L \rangle^2 = \frac{e^2}{8\pi^2} \int_{0^+}^{\infty} d\omega \, \omega \, \mathrm{Tr} \, \Sigma_L^\dagger(\omega) \Sigma_L(\omega),
\tag{5.4a}
$$

$$
\mathrm{var}(Q_R) \equiv \langle Q_R^2 \rangle - \langle Q_R \rangle^2 = \frac{e^2}{8\pi^2} \int_{0^+}^{\infty} d\omega \, \omega \, \mathrm{Tr} \, \Sigma_R^\dagger(\omega) \Sigma_R(\omega),
\tag{5.4b}
$$

$$
\begin{aligned}
\mathrm{covar}(Q_L Q_R) &\equiv \tfrac{1}{2} \langle Q_L Q_R \rangle + \tfrac{1}{2} \langle Q_R Q_L \rangle - \langle Q_L \rangle \langle Q_R \rangle \\
&= \frac{e^2}{16\pi^2} \int_{0^+}^{\infty} d\omega \, \omega \, \mathrm{Tr} \left[ \Sigma_L^\dagger(\omega) \Sigma_R(\omega) + \Sigma_R^\dagger(\omega) \Sigma_L(\omega) \right],
\end{aligned}
\tag{5.4c}
$$

in terms of the matrices

$$
\Sigma_L(\omega) = \int_{-\infty}^{\infty} dt \, e^{i\omega t} \Sigma_L(t), \quad \Sigma_L(t) = S_F^\dagger(t) \begin{pmatrix} \nu_y & 0 \\ 0 & 0 \end{pmatrix} S_F(t),
\tag{5.5a}
$$

$$
\Sigma_R(\omega) = \int_{-\infty}^{\infty} dt \, e^{i\omega t} \Sigma_R(t), \quad \Sigma_R(t) = S_F^\dagger(t) \begin{pmatrix} 0 & 0 \\ 0 & \nu_y \end{pmatrix} S_F(t).
\tag{5.5b}
$$

---

[4] A calculation along the lines of Ref. 15 of the average number of quasiparticles transferred per cycle into the left or the right lead gives $\langle N_L \rangle = \langle N_R \rangle = 42\zeta(3)/\pi^4 = 0.518$.

The lower limit $0^+$ in the $\omega$-integrals (5.4) avoids a spurious contribution $\propto \delta(\omega)$.

From the expression (4.1) for $S_{\mathrm{F}}(t)$ we find

$$
\begin{aligned}
\mathrm{Tr}\, \Sigma_{\mathrm{L}}^{\dagger}(\omega)\Sigma_{\mathrm{L}}(\omega) &= \mathrm{Tr}\, \Sigma_{\mathrm{R}}^{\dagger}(\omega)\Sigma_{\mathrm{R}}(\omega) \\
&= \tfrac{1}{2}|Z_+(\omega)|^2 + \tfrac{1}{2}|Z_+(-\omega)|^2 + \tfrac{1}{2}|Z_-(\omega)|^2 + \tfrac{1}{2}|Z_-(-\omega)|^2,
\end{aligned}
\tag{5.6a}
$$

$$
\begin{aligned}
\mathrm{Tr}\, \Sigma_{\mathrm{L}}^{\dagger}(\omega)\Sigma_{\mathrm{R}}(\omega) &= \mathrm{Tr}\, \Sigma_{\mathrm{R}}^{\dagger}(\omega)\Sigma_{\mathrm{L}}(\omega) \\
&= \tfrac{1}{2}|Z_+(\omega)|^2 + \tfrac{1}{2}|Z_+(-\omega)|^2 - \tfrac{1}{2}|Z_-(\omega)|^2 - \tfrac{1}{2}|Z_-(-\omega)|^2,
\end{aligned}
\tag{5.6b}
$$

$$
Z_{\pm}(\omega) = \int_{-\infty}^{\infty} dt\, e^{i\omega t} e^{i\alpha_1(t) \pm i\alpha_3(t)}.
\tag{5.6c}
$$

The dependence on $\alpha_2$ and $\alpha_4$ drops out.

Without further calculation we see that for $\alpha_1 = \alpha_3$ the contribution of $Z_-(\omega)$ to the correlators (5.4) vanishes, hence $\mathrm{covar}(Q_{\mathrm{L}}Q_{\mathrm{R}}) = \mathrm{var}(Q_{\mathrm{L}}) = \mathrm{var}(Q_{\mathrm{R}})$. This implies that the charge difference $Q_{\mathrm{L}} - Q_{\mathrm{R}}$ is zero without fluctuations,

$$
\mathrm{var}(Q_{\mathrm{L}} - Q_{\mathrm{R}}) = \mathrm{var}(Q_{\mathrm{L}}) + \mathrm{var}(Q_{\mathrm{R}}) - 2\,\mathrm{covar}(Q_{\mathrm{L}}Q_{\mathrm{R}}) = 0.
\tag{5.7}
$$

The charges $Q_{\mathrm{L}}$ and $Q_{\mathrm{R}}$ do fluctuate individually, with a variance close to $e^2/4$, and so does the sum $Q_{\mathrm{L}} + Q_{\mathrm{R}}$, with a variance close to $e^2$. These values can be calculated precisely for the time dependence [15]

$$
\alpha(t) \approx \arccos\left[ \tanh\left( \frac{W}{\xi_0} \frac{\pi - \phi(t)}{2} \right) \right] \approx \arccos[-\tanh(t/2t_\phi)],
\tag{5.8}
$$

which is an accurate representation of Eq. (4.2) for $W/\xi_0 \gg 1$. We find

$$
Z_+(\omega) = 2\pi\delta(\omega) - \frac{8\pi\omega t_\phi^2}{\sinh(\pi\omega t_\phi)} + \frac{8\pi\omega t_\phi^2}{\cosh(\pi\omega t_\phi)}, \quad Z_-(\omega) = 2\pi\delta(\omega),
\tag{5.9}
$$

$$
\Rightarrow \mathrm{var}(Q_{\mathrm{L}}) = \mathrm{var}(Q_{\mathrm{R}}) = \tfrac{1}{4}\mathrm{var}(Q_{\mathrm{L}} + Q_{\mathrm{R}}) = \frac{21\zeta(3)}{\pi^4} e^2 = 0.259\, e^2.
\tag{5.10}
$$

For $\alpha_1 \neq \alpha_3$ we can evaluate the integrals numerically using the time dependence

$$
\alpha_n = \arccos\left[ -\tanh(t/2t_n) \right],
\tag{5.11}
$$

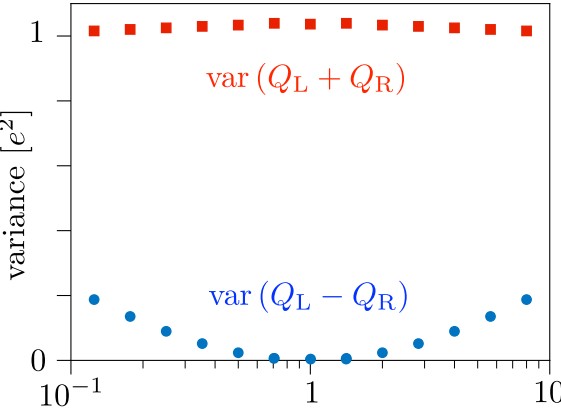

Figure 5: Variance of the sum and difference of the transferred charges upon fusion of the edge vortices in Josephson junctions $J_2$ and $J_4$, as a function of the asymmetry in the width of the injecting Josephson junctions $J_1$ and $J_3$.

increasing from 0 to $\pi$ in a time $t_n = (\xi_0/W_n)(\Delta t/2\pi)$ around $t = 0$. Results for $\mathrm{var}\,(Q_L \pm Q_R)$ are shown in Fig. 5. The shot noise for the charge difference remains suppressed for a moderately large deviation from unity of $W_1/W_3$.

# 6 Conclusion

We have shown how the method of time-resolved and "on-demand" injection of edge vortices proposed in Ref. 15 can be used to demonstrate the non-Abelian fusion rule of Majorana zero-modes. The signature of the correlated but non-deterministic outcome of the fusion of two pairs of edge vortices is a fluctuating electrical current $I_L$ and $I_R$ through two Josephson junctions, induced by a $2\pi$ phase shift of the pair potential. While the sum $I_L + I_R$ has average $e$ per cycle and variance close to $e^2$, the difference $I_L - I_R$ vanishes without fluctuations in a symmetric structure (and remains much below $e^2$ for moderate asymmetries).

The four-terminal structure of chiral Majorana edge modes that we have studied has been investigated before in the context of the injection of fermions [26–29]. A Majorana fermion that splits into partial waves at opposite edges defines a nonlocally encoded *charge qubit*: a coherent superposition of an electron and a hole.[5] In contrast, the injection of vortices at opposite edges is a nonlocal encoding of the *fermion parity*. The difference could be significant for quantum information processing if the fermion parity qubit is more robust against decoherence than the charge qubit. We surmise that zero-modes in edge vortices are better protected against charge noise and other local sources of decoherence than Majorana fermions — basically because a Majorana fermion is charge neutral on average but does exhibit quantum fluctuations of the charge.

Much further research is needed to substantiate the potential of edge vortices as carriers of quantum information, but we feel that they have much to offer at least for the demonstration of basic operations in topological quantum computation: the braiding operation of Ref. 15 and the non-deterministic fusion operation considered here.

# Acknowledgements

P. Baireuther suggested to us the vortex fusion geometry of Fig. 2. This research was supported by the Netherlands Organization for Scientific Research (NWO/OCW) and by the European Research Council (ERC).

# A Calculation of the frozen scattering matrix

Consider first the stationary scattering problem, when the four-terminal Josephson junction from Fig. 3 has a time-independent phase difference $\phi$. This gives the "frozen" scattering matrix $S_F(E, \phi)$, which we evaluate at the Fermi level ($E = 0$).

As calculated in Ref. 15, each of the four terminals (width $W_n$) has at the Fermi level a

---

[5]The splitting of a Majorana fermion into partial waves does not provide a local encoding of the fermion parity because a measurement at one edge can detect the presence or absence of a fermion.

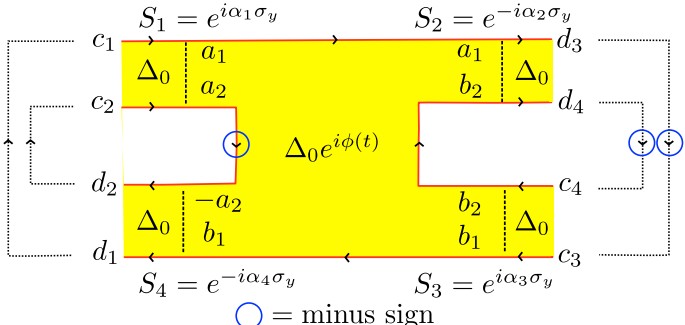

$\bigcirc$ = minus sign

Figure 6: Labeling of incoming and outgoing Majorana edge modes in a four-terminal Josephson junction.

scattering matrix in SO(2) given by

$$S_n = \begin{pmatrix} \cos\alpha_n & \sin\alpha_n \\ -\sin\alpha_n & \cos\alpha_n \end{pmatrix} = e^{i\alpha_n \nu_y} \text{ for } n = 1,3,$$

$$S_n = \begin{pmatrix} \cos\alpha_n & -\sin\alpha_n \\ \sin\alpha_n & \cos\alpha_n \end{pmatrix} = e^{-i\alpha_n \nu_y} \text{ for } n = 2,4. \tag{A.1}$$

The Pauli matrix $\nu_y$ acts on the two Majorana modes at a Josephson junction. The angles $\alpha_n$ are given as a function of $\phi$ and the ratio $W_n/\xi_0$ by Eq. (4.2) from the main text.

Referring to the labeling of modes from Fig. 6, we have the linear relations

$$\begin{pmatrix} d_1 \\ d_2 \\ d_3 \\ d_4 \end{pmatrix} = S_F \begin{pmatrix} c_1 \\ c_2 \\ c_3 \\ c_4 \end{pmatrix}, \tag{A.2a}$$

$$\begin{pmatrix} a_1 \\ a_2 \end{pmatrix} = S_1 \begin{pmatrix} c_1 \\ c_2 \end{pmatrix}, \quad \begin{pmatrix} d_3 \\ d_4 \end{pmatrix} = S_2 \begin{pmatrix} a_1 \\ b_2 \end{pmatrix}, \quad \begin{pmatrix} b_1 \\ b_2 \end{pmatrix} = S_3 \begin{pmatrix} c_3 \\ c_4 \end{pmatrix}, \quad \begin{pmatrix} d_1 \\ d_2 \end{pmatrix} = S_4 \begin{pmatrix} b_1 \\ -a_2 \end{pmatrix}. \tag{A.2b}$$

The minus sign for the coefficient $a_2$ in the last equality accounts for the $\pi$ Berry phase of a circulating Majorana edge mode. As indicated by the dotted lines in Fig. 6, the edge modes are segments of three closed loops. We choose a gauge where the minus sign in each loop is acquired on the downward branch, indicated by the blue circle. This only affects the branch with amplitude $a_2$, because the other two downward branches are outside of the scattering region.

Elimination of the $a_n$ and $b_n$ variables gives

$$S_F = \begin{pmatrix} -\sin\alpha_1 \sin\alpha_4 & \cos\alpha_1 \sin\alpha_4 & \cos\alpha_3 \cos\alpha_4 & \cos\alpha_4 \sin\alpha_3 \\ \cos\alpha_4 \sin\alpha_1 & -\cos\alpha_1 \cos\alpha_4 & \cos\alpha_3 \sin\alpha_4 & \sin\alpha_3 \sin\alpha_4 \\ \cos\alpha_1 \cos\alpha_2 & \cos\alpha_2 \sin\alpha_1 & \sin\alpha_2 \sin\alpha_3 & -\cos\alpha_3 \sin\alpha_2 \\ \cos\alpha_1 \sin\alpha_2 & \sin\alpha_1 \sin\alpha_2 & -\cos\alpha_2 \sin\alpha_3 & \cos\alpha_2 \cos\alpha_3, \end{pmatrix}, \tag{A.3}$$

which may be written more compactly as Eq. (4.1). One can check that $S_F \in \mathrm{SO}(4)$, in particular, it has determinant $+1$ as it should be in the absence of a Majorana zero-mode [38].[6]

In the adiabatic regime the scattering matrix $S(E, E')$ of the time-dependent problem is related to the frozen scattering matrix $S_F(E, \phi)$ via

$$S(E + \tfrac{1}{2}\omega, E - \tfrac{1}{2}\omega) \approx \int_{-\infty}^{\infty} dt \, e^{i\omega t} S_F(E, \phi(t)). \tag{A.4}$$

---

[6]If we would not have accounted for the sign change of $a_2$ the determinant of $S_F$ would have been $-1$.

Near the Fermi level we may furthermore neglect the dependence on the average energy, approximating

$$S(E, E') \approx \int_{-\infty}^{\infty} dt \, e^{i(E-E')t} S_{\mathrm{F}}(0, \phi(t)). \tag{A.5}$$

# B    Derivation of the Klich formula

The operator trace (3.19) for particle-hole conjugate Majorana operators $a(E) = a^{\dagger}(-E)$ can be derived from the Klich formula (3.13) for self-conjugate Majorana operators $\gamma = \gamma^{\dagger}$, by performing a unitary transformation:

$$\begin{pmatrix} \gamma_n(E) \\ \gamma'_n(E) \end{pmatrix} = U \begin{pmatrix} a_n(E) \\ a_n^{\dagger}(E) \end{pmatrix}, \quad U = \frac{1}{\sqrt{2}} \begin{pmatrix} 1 & 1 \\ -i & i \end{pmatrix}. \tag{B.1}$$

At positive energies the $\gamma$ operators satisfy the Clifford algebra of Majorana operators,

$$\{\gamma_n(E), \gamma_m(E')\} = \{\gamma'_n(E), \gamma'_m(E')\} = \{\gamma_n(E), \gamma'_m(E')\} = \delta_{nm}\delta_{EE'}, \;\; E, E' > 0. \tag{B.2}$$

Note that

$$\gamma_n(E)^2 = \gamma'_n(E)^2 = 1/2. \tag{B.3}$$

The bilinear form (3.15) of the $a$ operators transforms into

$$a^{\dagger} \cdot O \cdot a = \sum_{n,m} \sum_{E,E'>0} \begin{pmatrix} \gamma_n(E) \\ \gamma'_n(E) \end{pmatrix} \tilde{O}_{nm}(E, E') \begin{pmatrix} \gamma_m(E') \\ \gamma'_m(E') \end{pmatrix}, \tag{B.4}$$

with $\tilde{O} = UOU^{\dagger}$. Because only positive energies appear in Eq. (B.4), we may apply the anticommutator (B.2), which implies that the traceless symmetric part of $\tilde{O}$ drops out. Only the trace $\mathrm{Tr}\,\tilde{O} = \mathrm{Tr}\,O$ and the antisymmetric part $(\tilde{O} - \tilde{O}^{\mathrm{T}})/2$ contribute,

$$a^{\dagger} \cdot O \cdot a = \tfrac{1}{2}\gamma \cdot (\tilde{O} - \tilde{O}^{\mathrm{T}}) \cdot \gamma + \tfrac{1}{2} \mathrm{Tr}\,O. \tag{B.5}$$

After these preparations we can apply Klich's original formula [32],

$$\left[ \mathrm{Tr} \prod_k \exp(a^{\dagger} \cdot O_k \cdot a) \right]^2 = \exp\left( \sum_k \mathrm{Tr}\,O_k \right) \mathrm{Det}\left( 1 + \prod_k \exp(\tilde{O}_k - \tilde{O}_k^{\mathrm{T}}) \right). \tag{B.6}$$

Finally we invert the unitary transformation,

$$U^{\dagger}\tilde{O}U = O, \;\; U^{\dagger}\tilde{O}^{T}U = (U^{\mathrm{T}}U)^{\dagger}O^{\mathrm{T}}(U^{\mathrm{T}}U) = \sigma_x O^{\mathrm{T}} \sigma_x, \tag{B.7}$$

to arrive at

$$\left[ \mathrm{Tr} \prod_k \exp(a^{\dagger} \cdot O_k \cdot a) \right]^2 = \exp\left( \sum_k \mathrm{Tr}\,O_k \right) \mathrm{Det}\left( 1 + \prod_k \exp(O_k - \sigma_x O_k^{\mathrm{T}} \sigma_x) \right), \tag{B.8}$$

which is Eq. (3.19).

## C  Scattering formulas for charge correlators

### C.1  General expressions for first and second moments

Moments of the transferred charge in the left lead are given by the expectation value

$$\langle Q_{\mathrm{L}}^{p}\rangle = \left\langle \left(\boldsymbol{a}^{\dagger}\cdot\boldsymbol{Q}\cdot\boldsymbol{a}\right)^{p}\right\rangle, \ \ \boldsymbol{Q} = \boldsymbol{S}^{\dagger}\mathcal{P}_{\mathrm{L}}\mathcal{P}_{+}e\,\nu_{y}\boldsymbol{S}. \tag{C.1}$$

In comparison with the number operator (3.6) there is a matrix $e\,\nu_{y}$ which is the charge operator in the Majorana basis. (It would be $e\,\nu_{z}$ in the particle-hole basis.) The expectation value $\langle\cdots\rangle = \mathrm{Tr}(\rho_{\mathrm{eq}}\cdots)$ is with respect to an equilibrium distribution of the $\boldsymbol{a}$ operators, with density matrix (3.7).

Because of the Majorana commutator (3.14), we have both the usual type-I average

$$\langle a_{n}^{\dagger}(E)a_{m}(E')\rangle = \delta_{nm}\delta(E-E')f(E), \ \ f(E) = (1+e^{\beta E})^{-1}, \tag{C.2}$$

and the unusual type-II average

$$\langle a_{n}(E)a_{m}(E')\rangle = \delta_{nm}\delta(E+E')f(-E), \ \ f(-E) = 1-f(E). \tag{C.3}$$

Averages of strings of $a$ and $a^{\dagger}$ operators are obtained by summing over all pairwise averages of both types I and II, signed by the permutation.[7] We assume zero temperature, when $f(E) = \mathcal{P}_{-}$ and $1-f(E) = \mathcal{P}_{+}$ are step functions of energy.

The first moment of the transferred charge contains a single type-I average,

$$\langle Q_{\mathrm{L}}\rangle = \mathrm{Tr}\,\mathcal{P}_{-}\boldsymbol{Q} = \int_{0}^{\infty}\frac{dE}{2\pi}\int_{-\infty}^{0}\frac{dE'}{2\pi}\,\mathrm{Tr}\,\boldsymbol{S}^{\dagger}(E,E')e\,\nu_{y}\mathcal{P}_{\mathrm{L}}\boldsymbol{S}(E,E'). \tag{C.4}$$

The variance contains a term with two type-I averages and a term with two type-II averages,

$$\mathrm{var}(Q_{\mathrm{L}}) = \mathrm{Tr}\,\mathcal{P}_{-}\boldsymbol{Q}\mathcal{P}_{+}\boldsymbol{Q} - \int_{0}^{\infty}\frac{dE}{2\pi}\int_{-\infty}^{0}\frac{dE'}{2\pi}\sum_{n,m}Q_{nm}(-E,-E')Q_{nm}(E,E'). \tag{C.5}$$

The particle-hole symmetry relation (3.5) of the scattering matrix implies that

$$Q_{nm}(-E,-E') = -(\boldsymbol{S}^{\dagger}\mathcal{P}_{\mathrm{L}}\mathcal{P}_{-}e\,\nu_{y}\boldsymbol{S})_{mn}(E',E). \tag{C.6}$$

Substitution into Eq. (C.5) gives

$$\mathrm{var}(Q_{\mathrm{L}}) = \mathrm{Tr}\,\mathcal{P}_{-}\boldsymbol{Q}\mathcal{P}_{+}\boldsymbol{Q} + \mathrm{Tr}\,\mathcal{P}_{-}\boldsymbol{Q}'\mathcal{P}_{+}\boldsymbol{Q}, \tag{C.7}$$

with $\boldsymbol{Q}'$ as in Eq. (C.1) upon replacement of $\mathcal{P}_{+}$ by $\mathcal{P}_{-}$. Since $\mathcal{P}_{+}+\mathcal{P}_{+} = 1$, this reduces to

$$\mathrm{var}(Q_{\mathrm{L}}) = \mathrm{Tr}\,\mathcal{P}_{-}(\boldsymbol{S}^{\dagger}\mathcal{P}_{\mathrm{L}}e\,\nu_{y}\boldsymbol{S})\mathcal{P}_{+}(\boldsymbol{S}^{\dagger}\mathcal{P}_{\mathrm{L}}\mathcal{P}_{+}e\,\nu_{y}\boldsymbol{S}). \tag{C.8}$$

It is convenient to eliminate the second $\mathcal{P}_{+}$ projector from Eq. (C.8). This can be done via particle-hole symmetry, which implies that

$$\begin{aligned}
\mathrm{Tr}\,\mathcal{P}_{-}(\boldsymbol{S}^{\dagger}\mathcal{P}_{\mathrm{L}}e\,\nu_{y}\boldsymbol{S})\mathcal{P}_{+}(\boldsymbol{S}^{\dagger}\mathcal{P}_{\mathrm{L}}\mathcal{P}_{+}e\,\nu_{y}\boldsymbol{S}) &= \mathrm{Tr}\,(\boldsymbol{S}^{\dagger}\mathcal{P}_{\mathrm{L}}\mathcal{P}_{+}e\,\nu_{y}\boldsymbol{S})^{\mathrm{T}}\mathcal{P}_{+}(\boldsymbol{S}^{\dagger}\mathcal{P}_{\mathrm{L}}e\,\nu_{y}\boldsymbol{S})^{\mathrm{T}}\mathcal{P}_{-} \\
&= \mathrm{Tr}\,(\boldsymbol{S}^{\dagger}\mathcal{P}_{\mathrm{L}}\mathcal{P}_{-}e\,\nu_{y}\boldsymbol{S})\mathcal{P}_{-}(\boldsymbol{S}^{\dagger}\mathcal{P}_{\mathrm{L}}e\,\nu_{y}\boldsymbol{S})\mathcal{P}_{+} \\
&= \mathrm{Tr}\,\mathcal{P}_{-}(\boldsymbol{S}^{\dagger}\mathcal{P}_{\mathrm{L}}e\,\nu_{y}\boldsymbol{S})\mathcal{P}_{+}(\boldsymbol{S}^{\dagger}\mathcal{P}_{\mathrm{L}}\mathcal{P}_{-}e\,\nu_{y}\boldsymbol{S}).
\end{aligned} \tag{C.9}$$

---

[7]An equivalent procedure [33] is to first use the relation $a_{n}(-E) = a_{n}^{\dagger}(E)$ to rewrite the expectation value such that only positive energies appear, and then apply Wick's theorem as usual.

Hence

$$\tfrac{1}{2}\mathrm{Tr}\,\mathcal{P}_-(\boldsymbol{S}^\dagger \mathcal{P}_\mathrm{L} e\, v_y \boldsymbol{S})\mathcal{P}_+(\boldsymbol{S}^\dagger \mathcal{P}_\mathrm{L}(\mathcal{P}_- - \mathcal{P}_+)e\, v_y \boldsymbol{S}) = 0, \tag{C.10}$$

and adding this to Eq. (C.8) we arrive at

$$\begin{aligned}
\mathrm{var}\,(Q_\mathrm{L}) &= \frac{1}{2}\mathrm{Tr}\,\mathcal{P}_-(\boldsymbol{S}^\dagger \mathcal{P}_\mathrm{L} e\, v_y \boldsymbol{S})\mathcal{P}_+(\boldsymbol{S}^\dagger \mathcal{P}_\mathrm{L} e\, v_y \boldsymbol{S}) \\
&= \frac{1}{2}e^2 \int_0^\infty \frac{dE}{2\pi} \int_{-\infty}^0 \frac{dE'}{2\pi}\,\mathrm{Tr}\,\Sigma_\mathrm{L}^\dagger(E,E',)\Sigma_\mathrm{L}(E,E'),\ \ \Sigma_\mathrm{L} = \boldsymbol{S}^\dagger \mathcal{P}_\mathrm{L} v_y \boldsymbol{S}.
\end{aligned} \tag{C.11}$$

The expressions for the other correlators are analogous,

$$\mathrm{var}\,(Q_\mathrm{R}) = \frac{1}{2}e^2 \int_0^\infty \frac{dE}{2\pi} \int_{-\infty}^0 \frac{dE'}{2\pi}\,\mathrm{Tr}\,\Sigma_\mathrm{R}^\dagger(E,E')\Sigma_\mathrm{R}(E,E'),\ \ \Sigma_\mathrm{R} = \boldsymbol{S}^\dagger \mathcal{P}_\mathrm{R} v_y \boldsymbol{S}, \tag{C.12}$$

$$\mathrm{covar}\,(Q_\mathrm{L}Q_\mathrm{R}) = \frac{1}{4}e^2 \int_0^\infty \frac{dE}{2\pi} \int_{-\infty}^0 \frac{dE'}{2\pi}\,\mathrm{Tr}\,\big[\Sigma_\mathrm{L}^\dagger(E,E')\Sigma_\mathrm{R}(E,E') + \Sigma_\mathrm{R}^\dagger(E,E')\Sigma_\mathrm{L}(E,E')\big]. \tag{C.13}$$

Eq. (C.13) gives the symmetrized covariance,

$$\mathrm{covar}(Q_\mathrm{L}Q_\mathrm{R}) \equiv \tfrac{1}{2}\langle Q_\mathrm{L}Q_\mathrm{R}\rangle + \tfrac{1}{2}\langle Q_\mathrm{R}Q_\mathrm{L}\rangle - \langle Q_\mathrm{L}\rangle\langle Q_\mathrm{R}\rangle, \tag{C.14}$$

appropriate for a calculation of $\mathrm{var}\,(Q_\mathrm{L} \pm Q_\mathrm{R})$.

## C.2  Adiabatic approximation

The general expressions (C.4) and (C.11)–(C.13) can be simplified in the adiabatic regime, when near the Fermi level $S(E,E')$ depends only on the energy difference $\omega = E - E'$. We use the identity

$$\int_0^\infty dE \int_{-\infty}^0 dE'\, F(E - E') = \int_{0^+}^\infty d\omega\, \omega F(\omega). \tag{C.15}$$

The lower integration limit $0^+$ eliminates a possibly singular delta function in $F(\omega)$, which should not enter in the excitation spectrum.

For the average transferred charge (C.4) we thus have

$$\langle Q_\mathrm{L}\rangle = \frac{1}{4\pi^2} \int_{0^+}^\infty d\omega\, \omega\, \mathrm{Tr}\, S^\dagger(\omega)e\, v_y \mathcal{P}_\mathrm{L} S(\omega). \tag{C.16}$$

As explained in Ref. 15, this is equivalent to the Brouwer formula (5.1): Because of

$$[S^\dagger(\omega)v_y \mathcal{P}_\mathrm{L} S(\omega)]^\mathrm{T} = -S^\dagger(-\omega)v_y \mathcal{P}_\mathrm{L} S(-\omega) \tag{C.17}$$

the integrand in Eq. (C.16) is an even function of $\omega$, hence the integration can be extended to $\int_{-\infty}^\infty d\omega$, and then transformation to the time domain gives Eq. (5.1).

For the second moments we use that the kernels $\Sigma(E,E') \mapsto \Sigma(\omega)$ are functions of $\omega = E - E'$ when $S(E,E') \mapsto S(\omega)$,

$$\begin{aligned}
\Sigma_\mathrm{L,R}(E,E') &= \int_{-\infty}^\infty \frac{dE''}{2\pi} S^\dagger(E'',E)\mathcal{P}_\mathrm{L,R} v_y S(E'',E') \\
\Rightarrow \Sigma_\mathrm{L,R}(\omega) &= \int_{-\infty}^\infty \frac{d\omega'}{2\pi} S^\dagger(\omega' - \omega)\mathcal{P}_\mathrm{L,R} v_y S(\omega') = \int_{-\infty}^\infty dt\, e^{i\omega t} S^\dagger(t)\mathcal{P}_\mathrm{L,R} v_y S(t).
\end{aligned} \tag{C.18}$$

The Fourier transform is defined as

$$S(\omega) = \int_{-\infty}^{\infty} dt\, e^{i\omega t} S(t). \tag{C.19}$$

Note that for the representation (C.18) of $\Sigma(\omega)$ as a single time integral it was essential that we eliminated the $\mathcal{P}_+$ projector from the scattering matrix product.

Application of Eqs. (C.15) and (C.18) to Eqs. (C.11)–(C.13) then gives the formulas (5.4) from the main text.

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
