# Peer review of "Electrical detection of the Majorana fusion rule for chiral edge vortices in a topological superconductor"

_SciPost Physics, doi:SciPost Phys. 6, 022 (2019)_

## Round 1 · Referee Report · Anonymous (Referee 1) · 2019-1-17

Strengths

Highly original. Opens a new line of research that might turn out to be a way to demonstration nonabelian behavior og Majorana zero-modes.

Weaknesses

The paper does not address possible error mechanisms or give estimates for the limiting time-scales.

Report

The paper is an important paper which suggest a new type of experiments using edge modes in hybrid topological insulator/superconductor systems. The type of experiment suggested in the paper is a natural next step in the rapidly developing field of topological superconductors and the search for reliable ways to investigate the nonabelian nature of their edge modes.

Requested changes

No requested changes.

---

## Round 1 · Referee Report · Anonymous (Referee 2) · 2019-2-3

Strengths

  1. The authors provide a novel approach aimed toward the demonstration of non-Abelian fusion rules of Majorana fermions in setups that may be in experimental reach in the foreseeable future. In contrast to most previous approaches where localized Majorana bound states have been considered, the authors adopt a novel viewpoint and study edge vortices in systems with chiral Majorana edges. In such a system the injection of vortices at opposite ends amounts to a nonlocal encoding of fermion parity. This idea may have certain advantages over other proposals but of course may also turn out to have disadvantages on its own. Nevertheless, I believe that in view of the current status of the field (where even basic Majorana fusion experiments are missing, or the very existence of Majorana bound states remains hotly debated) a new promising proposal such as the one presented here is more than welcome.
  2. The paper is nicely written and fun to read. Even people not working directly in the field can understand it.
  3. Methodologically, the scattering approach and the Klich formula are here considered for Majorana systems. The results can then be obtained by analysis of Toeplitz determinants, and allow even for analytical results.
  4. Detection of fusion could be possible in their scheme by measuring currents and their shot noise. This simplicity is a big advantage.

Weaknesses

  1. I only have one minor point: when citing [10-13] I was missing the important paper by Flensberg and collaborators in New J. Phys. (2017), which is very closely related to [12,13] and should be cited together with those. Other than that, the references are rather complete and adequate.

Report

The paper deserves publication once the above point has been taken care of.

Requested changes

see above

---

## Editorial Decision

published